# Corrosion Properties of Boron- and Manganese-Alloyed Stainless Steels as a Material for the Bipolar Plates of PEM Fuel Cells

**DOI:** 10.3390/ma15196557

**Published:** 2022-09-21

**Authors:** Tomáš Lovaši, Vojtěch Pečinka, Jakub Ludvík, Jiří Kubásek, Filip Průša, Milan Kouřil

**Affiliations:** Department of Metals and Corrosion Engineering, University of Chemistry and Technology, Prague, Technická 5, 166 28 Prague, Czech Republic

**Keywords:** stainless steels, corrosion, bipolar plates

## Abstract

Stainless steels are materials that could be used for constructing not only the bearing parts of fuel cells but also the functional ones, particularly the bipolar plates. The advantage of stainless steel is its valuable electrical and thermal conductivity, reasonably low cost, excellent mechanical properties, and good formability. Paradoxically, the self-protection effect resulting from passivation turns into the main disadvantage, which is unacceptable interfacial contact resistance. The aim of this study was to test a number of possible stainless steels in a simulated fuel cell environment, especially those alloyed with boron and manganese, which were found to improve the contact resistance properties of stainless steels. The primary focus of the study is to determine the corrosion resistance of the individual materials tested. Electrochemical tests and contact resistance measurements were performed following the DOE requirements. Manganese-alloyed LDX stainless steel achieved the best results in the electrochemical tests; the worst were achieved by boron-containing steels. Boron-containing stainless steels suffered from localized corrosion resulting from chromium-rich boride formation. All steels tested exceeded the DOE limit in the contact resistance measurement, with 316L reaching the lowest values.

## 1. Introduction

Hydrogen fuel cells with proton exchange membranes (PEMFCs), which transform chemical energy into electrical energy, have been considered as a substitute for combustion engines in the near future [1,2]. The fuel cell itself is supposed to have many advantages in comparison to conventional combustion engines. The main advantages are, for instance, the reduced production of emissions, such as CO_2_, NO_x_, and SO_x_, reduction in noise, etc. Immediate responses to changes in the desired power and higher energy efficiency can be also mentioned among the advantages [3,4,5]. Owing to such qualities, fuel cells represent a promising energy technology for the future. However, there are also some disadvantages that prevent the wider application of fuel cells in practice at present. The main disadvantages are economic challenges related to their production. Demands on weight and volume following automotive industry requirements [6] are challenging as well. The price of a fuel cell depends on the materials used in the manufacture of its individual components. Bipolar plates form a significant portion of the price. Bipolar plates comprise up to 80% of the total weight and 45% of the total price of a fuel cell [7,8].

PEMFCs are composed of several different parts, including a gas diffusion layer and a catalyst layer (MEA—membrane electrode assembly), which are situated between two bipolar plates. The membrane forms a solid electrolyte in PEMFCs. The most commonly used membrane—perfluorosulfonic acidobasic polymer (PFSA)—has a main chain formed of PTFE and side chains ending in sulfonic acidic groups [9,10]. Bipolar plates used to be made of graphite because of its electrical properties, chemical stability, and corrosion resistance. The disadvantage of graphite is its brittleness and porosity. These disadvantages are associated with machining problems, which are therefore expensive and time-consuming [7,11]. The use of graphite composites, mixtures of polymers (e.g., polypropylene), and graphite particles appears to be an alternative to pure graphite. Although the polymer matrix improves the mechanical properties of the bipolar plate, on the other hand, it reduces the electrical conductivity [7,11,12,13]. Since the bipolar plate material must meet the requirements according to the current US Department of Energy (DOE) limits, it is necessary to look for suitable materials, for example, among metals. Therefore, the current research focuses on the use of metallic materials with conductive coatings and anticorrosion properties [14].

The main candidates from nonferrous metals are titanium [15,16,17,18,19], copper [20,21,22,23], magnesium [24,25], aluminum [26,27,28,29], and their alloys. A disadvantage of these materials is their higher price related to improving their corrosion resistance in the environment of PEMFCs by various coatings. Another substitute under consideration, which is being extensively tested, is stainless steel. This material seems to be very promising due to its properties, such as electrical and thermal conductivity, and mechanical properties. In addition, it is affordable and relatively easy to machine. On the other hand, stainless steels are prone to corrosion and the formation of localized corrosion attacks in the environment where the fuel cells are supposed to operate [30,31,32]. The corrosion of the materials can damage the polymeric membrane by releasing ions into the environment. Moreover, it can increase the contact resistance depending on the thickness of a passive layer, which is created on the surface of the steel. However, corrosion resistance can be improved by various coatings (titanium dioxide, nitrides, carbon) and combined coatings [33,34,35,36]. Another disadvantage of stainless steels is their small forming elongation. This problem is mainly associated with the stamping or hydroforming manufacturing process. In the final production, a shape error caused by springing may occur [37]. Therefore, it is necessary that the material from which the bipolar plate will be formed has the required elongation. Table 1 summarizes the most commonly used stainless steels as possible materials for making bipolar plates, their ductility, and the required ductility according to the DOE. Of the stainless steels, only some steels meet this requirement. In most cases, these are austenitic corrosion-resistant steels. Lower elongation values can be observed for duplex stainless steels compared to austenitic stainless steels. However, this problem could be eliminated by a different production method. One option is a combination of laser additive manufacturing and conventional machining [38,39].

The most attention is paid to austenitic stainless steels, particularly AISI 316L [43,44,45]. Austenitic stainless steels 304, 310, and 316 are tested to a lesser extent. Super-austenitic stainless steels, such as 254SMO and 904L, or duplex stainless steels, such as 2205, also come into consideration [11,46,47,48,49,50,51,52,53,54]. Boron-containing steels might be promising from the acceptable contact resistance point of view [55], as the transition metal borides offer a combination of high electrical conductivity and good corrosion resistance that make them attractive for potential use as a protective surface layer in this application [56]. Iversen [57] identified a beneficial effect of manganese’s addition to decrease the electrical resistance of stainless steels. Low-cost manganese-containing duplex stainless steel has been involved in the study for verifying the beneficial contribution of manganese. With regard to the large number of requirements that fuel cells have to meet, the DOE limits were created in order to ensure a sufficient lifespan and the necessary power [58]. The DOE requirements are summarised in Table 2.

The aim of this study was to test selected stainless steels and compare them on the basis of the results obtained in each test. Each of these tests was designed to verify the suitability of using the steel as a material for bipolar plates from a corrosion properties point of view.

## 2. Materials and Methods

### 2.1. Materials and Model Solution

Prior to each measurement, a fresh model solution was prepared according to DOE requirements. The solution consisted of 1 mL of 0.01% hydrofluoric acid and was topped up to 1 L with distilled water. The pH of the model solution was adjusted to 3 using 1 mol L^−1^ sulfuric acid. In this study, five stainless steels were tested, namely 316L, Fe-B, Fe-B-PM, LDX, and 2205. All steels were prepared by the classical metallurgical process, i.e., casting and post-processing, with the exception of Fe-B-PM steel. This steel was prepared by powder metallurgy (PM) using powdered ferroalloys. The formation of a eutectic mixture (Me_2_B-based) occurs only in the metallurgical process. The chemical composition of the individual stainless steels was determined by X-ray fluorescence spectroscopy (XRF) and optical emission spectroscopy (OES) at several locations and is summarized in Table 3. Samples were taken from the steels to observe their microstructures. The samples were ground on SiC grinding paper P80–P2500 and then polished using diamond paste with a particle size of 3/2 µm. The prepared samples were etched to reveal the microstructures. The boron-containing steels were chemically etched for 10 s in a mixture of 50 mL of nitric acid, 25 mL of hydrochloric acid, and 25 mL of distilled water. The other steels were electrochemically etched in 10% oxalic acid for 30–90 s. The microstructures of the steels were observed on an OLYMPUS PME-3 light metallographic microscope and a TESCAN VEGA 3 LMU scanning electron microscope equipped with an OXFORD Instruments INCA 350 dispersive spectrometer (SEM-EDS). The phase composition was determined by X-ray diffraction (PANanalyticalX’Pert Pro, radiation Co Kα).

### 2.2. Electrochemical Measurements

For the electrochemical measurements, a traditional three-electrode arrangement in a corrosion cell was used (Figure 1). The model solution was warmed up in the corrosion cell using a thermostat to 80 °C throughout the electrochemical measurements.

The steel sample was used as the working electrode, while the platinum wire was used as the counter electrode. A saturated silver chloride reference electrode (ACLE) was used as the reference electrode. A Zahner Zennium potentiostat was used to perform the individual electrochemical measurements. Thales XT Analysis software was used to analyze and process the collected data. Each measurement was repeated at least twice to confirm the reproducibility of the measured data. Electrochemical measurements were divided into three categories according to DOE requirements: accelerated corrosion tests, potentiodynamic tests, and potentiostatic tests. Samples for electrochemical measurements were prepared in the form of flat cylinders with a diameter of 16 mm. The exposed area was 1.02 cm^2^. Before exposure, the samples were always ground with P1200 emery paper to remove the passive layer from the surface. The aim was to prevent the spontaneous passivation of the sample surface prior to exposure.

### 2.3. Accelerated Corrosion Tests

The measurement of the free corrosion potential (E_corr_) was carried out for 1800 s and was followed immediately by the polarization resistance (R_p_) scan. The linear polarization ranged from −20 mV vs. E_corr_ to +20 mV vs. E_corr_ with a scan rate of 0.1 mV s^−1^. Polarization resistance values for individual steels were set within ±10 mV vs. E_corr_. The measurement of polarization resistance was followed by the measurement of potentiodynamic curves. The cathodic part of the polarization curve was measured in the potential range of +100 mV vs. E_corr_ to −1.2 V vs. E_ref_ with a scan rate of 5 mV s^−1^. The anodic portion of the curve was measured at the same scan rate over a potential range of −100 mV vs. E_corr_ to 1.5 V vs. E_ref_. All tests were measured in an aerated solution.

### 2.4. Potentiodynamic Polarization According to DOE Requirements

The DOE requirements for anode corrosion rate were set to be less than 1 µA cm^−2^ and without the occurrence of any active peak. Prior to the potentiodynamic measurements, E_corr_ and R_p_ measurements were made under conditions corresponding to accelerated corrosion tests. Potentiodynamic measurements were performed over a potential range of −0.4 V to 0.6 V vs. ACLE with a scan rate of 0.1 mV s^−1^. The model solution was purged with nitrogen throughout the measurement.

### 2.5. Potentiostatic Polarization According to DOE Requirements

The DOE requirements for cathode corrosion rate were also set to be less than 1 µA cm^−2^. Unlike the anode, the current response of the cathode was monitored under potentiostatic polarization at 0.6 V vs. ACLE over 24 h. In this case, the model solution was aerated, which simulates the cathodic-site conditions in fuel cells.

### 2.6. Interfacial Contact Resistance (ICR)

In addition to the corrosion tests, contact resistance measurements were performed on each type of steel tested. Contact resistance measurements were performed both before and after the potentiostatic test. According to DOE requirements, the areal specific resistance should not exceed 0.01 Ω cm^2^ under a load of 138 N cm^−2^. Sample preparation was the same as before corrosion tests. The contact resistance was measured between the sample surface and the carbon paper, as shown in Figure 2.

Sigracet 38 BC carbon paper from SGL Carbon was sandwiched between the surface of the sample and the copper cylinders. A universal testing machine FPZ 100/1 with gradually increasing force (0.3 mm min^−1^) was used to apply the compressive force between the copper cylinders and to record it. The copper cylinders were connected using clamps to a GOM-805 DC Milliohmeter, which recorded the resistance of the whole system (R_total_) by means of a four-wire technique. The contact resistance was then calculated using Equation (1) [60,61]:R_S/CP_ = 0.5(R_total_ − R_CP/CU_)(1)
where R_S/CP_ is the resistance at the sample–carbon paper interface and R_CP/Cu_ is the resistance at the carbon paper–copper cylinder interface. As a reference measurement for calculating the contact resistance between the sample and the carbon paper (R_S/CP_), a system without a sample was tested under the same conditions. Assuming that the resistances of the diffusion layer and the sample are small compared to the other values, they can be neglected and the contact resistance between the sample and the diffusion layer can be calculated.

## 3. Results and Discussion

### 3.1. Microstructure of Tested Stainless Steels

Figure 3 shows the scanning electron microscope images for each of the stainless steels tested. A fine-grained structure (grain size 10–40 µm) was observed in all materials, consisting mainly of γ-iron—austenite (steel 316L). The 2205 and LDX steels are two-phase, with the areas protruding to the surface being austenitic grains and the surrounding areas being ferrite grains. The electrochemical etching of these steels resulted in the selective etching of the less noble phase, which was confirmed by subsequent point chemical analysis. Figure 4 shows the results of X-ray diffraction. This method was used to confirm the phase composition of the tested materials. For the two-phase steels, the ratio of the fcc phase to the bcc phase is approximately 50% to 50% (LDX 45% bcc, 55% fcc; steel 2205 40% bcc, 60% fcc).

In both boron-containing steels, it can be observed that the formed particles are mainly enriched with Cr and Mn, as shown in Figure 5. As reported by Stoulil et al. [62], a eutectic system based on (Cr, Fe)_2_B is formed during the manufacturing process of these materials. The formation of a (Fe, Cr)_2_B-based eutectic system also follows from the binary diagrams of the individual components at a particular composition [63]. The increasing boron content in stainless steels causes the selective segregation of chromium in the structure. Boron bound to chromium forms (Fe, Cr)_2_B-type borides and M_23_(C, B)_6_ carbide-borides, which do not participate in the formation of the passive layer. The X-ray diffraction result (Figure 4) suggests the formation of Cr_2_B. The orthorhombic crystal lattice of the Cr_2_B phase may contain a proportion of iron and residual alloying elements. Therefore, this boride can be presented as Me_2_B, where Me = Fe, Cr, Mn [64,65,66]. The distribution of boride particles seems to be more favorable for Fe-B (PM) steel (compared to Fe-B steel). The particles are globular in shape and approximately 5 µm in size. The particles are free of any stretching and are uniformly distributed in the matrix. On the other hand, Fe-B steel exhibits an apparently heterogeneous structure, which is formed by stretched boride particles/eutectics and their size ranges from 2 to 100 µm. The difference in the microstructure of these steels is due to a different preparation process. The disadvantage of this steel is its chromium depletion of the surrounding matrix from the original 20–19 wt.% to 14 wt.%, which was observed by linear scan analysis. Chromium depletion in the matrix was observed in all volumes of the material, resulting in reduced resistance to localized corrosion. It was found by spot chemical analysis that areas or particles with higher Cr content (approximately 55 wt.%) also contained significant amounts of B compared to the matrix where B was not quantified. As stated in [67], as the boron content increases, the chromium content in the matrix decreases, which was also confirmed. For the boron-containing steels, the areal distribution of the particles was analyzed using ImageJ software. It has been shown that most of the surface area in Fe-B (PM) steels is covered by ineffective particles in terms of passivation. From this, we can conclude that most of the area is depleted of chromium. It can also be assumed that the corrosion resistance of these materials decreases significantly due to the formation of these eutectics. On the other hand, the formation of borides can manifest itself in a positive point of view. Since most of the surface is covered with boride particles, it can be assumed that a thick passive layer, which would cover the entire surface and would increase the value of the contact resistance, is not formed.

### 3.2. Accelerated Corrosion Tests

The average values of the free corrosion potentials (E_corr_) recorded during all measurements for each material are summarised in Table 4. The E_corr_ values were determined as the average of the last values after the free corrosion potential measurements were completed. The values of the free corrosion potential do not vary significantly from material to material. The potentials of the individual materials indicate their passive state.

As mentioned in the experimental section, polarization resistance was measured not only in accelerated corrosion tests but also before potentiodynamic and potentiostatic polarization. The values of all tested materials are shown in Figure 6. The highest values were measured for steels 316L, 2205, and LDX, but the latter steel shows significant deviations, which may be related to the insufficient protection of the passive layer. The Fe-B and Fe-B (PM) steels achieved very similar polarization resistance values. The lower polarization resistance values compared to the other samples could be related to the higher boron content, which binds more of the chromium, leading to the preferential formation of chromium borides. Consequently, the depletion of chromium in the matrix resulted in the formation of a passive layer on the steel surface with fewer protective properties. The polarization resistance value provides non-destructively quick information on the corrosion resistance of the tested steels in the PEMFC model environment. A relative comparison of the corrosion rates (v_corr_) of the individual steels follows from relation (2):v_corr_ ≈ 1/R_p_(2)

After free corrosion potential and polarization resistance measurements, potentiodynamic curves were measured to compare the materials and their resistance in the PEMFC. The anodic and cathodic sections of the curves were measured separately to avoid affecting the results with irreversible changes in condition at the working electrode surface resulting from the polarization. The cathodic parts of the potentiodynamic curves are shown in Figure 7. Since it is possible to find a large number of publications on 316L steel, this steel was used as a reference material [45,68] and all other materials tested were compared with this steel. The measurement itself is affected by many parameters, such as the mixing of the electrolyte and its temperature. This results in the formation of bubbles on the surface of the reference and working electrodes. During the measurement, these bubbles detach from the surface, causing a change in the exposed area and the measured current, resulting in the oscillation of the measured current. Another parameter that changes during the measurement is the oxygenation of the electrolyte, as a new electrolyte is prepared before each measurement. Figure 7 shows that 316L steel exhibits the most positive E_corr_ compared to the other steels tested. Small changes in the corrosion current density can also be observed. In addition, a small undulation of the curves can be observed (in the case of 316L and 2205). This may be due to the reduction in the environment itself (limiting the region of O_2_ reduction in the range of −0.1 to −0.3 V vs. ACLE, and for H_2_ approximately −0.45 V vs. ACLE). In the case of the other steels, there was an E_corr_ shift, which may be due to the material itself, as E_corr_ shifts to negative values as the chromium content of stainless steels increases. The determination of a cathodic Tafel slope (β_C_) is not straightforward in this case because there is no linear charge transfer region on the curves in semi-logarithmic coordinates. The curves have a linear region only in the limiting oxygen control region, so it is not possible to unambiguously determine β_C_ for all the tested materials. Therefore, the values given in Table 5 and Table 6 are indicative only.

The anodic sections of the polarization curves for steels in a PEMFC environment are shown in Figure 8. All steels tested were again compared to 316L steel. It can be observed that all materials are passivated, although the corrosion current density is higher than that of a conventional passivated material. For steels 316L, 2205, and LDX, a decrease in current density can be observed at a potential of approximately +1 V relative to ACLE, with a recurring increase in current density with the progressive polarization and increasing oxidation ability of the environment. This phenomenon may be related to transformations of Fe(II) to Fe(III) in the passive layer. By further increasing the potential, the electrolyte decomposes to form O_2_ at the working electrode surface. In the case of Fe-B and Fe-B (PM) steels, the decrease in current was not observed. As discussed in Section 3.1, the addition of boron has been shown to cause the formation of boride particles in the microstructure, while the matrix is depleted of chromium. In the case of 2205 steel, it can be observed that the material shows signs of activity at the onset of polarization. By further increasing the potential, it enters a passive state. This may be induced by the activation of the less noble ferrite phase in this two-phase steel since the ferrite is less resistant in this case.

### 3.3. Potentiodynamic Polarization According to DOE

Potentiodynamic polarization tests were also performed on the tested materials according to predetermined DOE requirements. The test results for each steel are shown in Figure 9. According to DOE regulations, all materials tested in corrosion on the anode should meet the requirement of <1 µA cm^−2^ and no active peak. During the measurements, there was no active peak at which the material actively dissolved for any of the materials tested. The value of 1 µA cm^−2^ was only gently exceeded at positive polarization potentials for steels 316L, Fe-B, and Fe-B (PM) (approximately 0.45 V versus ACLE).

The highest current density values, and hence the highest dissolution rates in the passive state, were repeatedly recorded for the boron-bearing stainless steels. In the literature, we can find results when 316L steel showed poor resistance to corrosion attacks [69,70,71,72]. On the other hand, the lowest dissolution rates in the passive state were recorded for the two-phase steels 2205 and LDX (Figure 10). Two-phase steels have the highest chromium content among the steels tested, which contributes to the formation of the passive layer. Likewise, the increased manganese content and the small addition of copper in LDX steels may have had a positive effect. In the study, the author points out the positive effect of manganese in this steel [57]. It is the addition of copper to LDX steel that increases the resistance of austenitic stainless steel to non-oxidizing acids. In this way, an increase in the passivation ability in a sulphuric and hydrofluoric acid environment is achieved [73,74]. Although Fe-B and Fe-B (PM) steels are highly alloyed with chromium and nickel, the negative influence of boron was confirmed, which was already evident in the polarization resistance measurements. However, it can be noted that there was a fairly significant scatter in the measured current densities for the individual materials tested. The current density is slightly above 1 µA cm^−2^ for steels 316L, Fe-B, and Fe-B (PM) at a highly oxidative range of the potentiodynamic polarization. These steels achieved almost the same current density values. LDX steel was the only steel tested that met the DOE limit in both the potentiodynamic scans. This steel performed well in the accelerated corrosion tests. Therefore, this steel can be considered promising. Steel 2205 can be described as partially compliant.

### 3.4. Potentiostatic Polarisation According to DOE

The last corrosion test was potentiostatic polarization at a potential of 0.6 V vs. ACLE for 24 h. The electrolyte was aerated during the measurements because the potentiostatic test simulates the cathodic conditions of the fuel cell with an excess of oxygen. As shown in Figure 11, fluctuations in the measured current values were observed. Small fluctuations in current in the case of 316L steel and LDX steel can be attributed to metastable local passive layer disturbance. With further polarization, the passive layer recovered and healed. Feng et al. observed [43] the effect of changing pH of the PEMFC environment on 316L steel corrosion behavior. As the pH decreased, the steel became passivated more easily down to pH 3. At this value, an oscillation of the measured current was observed, indicating the instability of the passive film. This phenomenon also applies to 2205 steel, to some extent. In the case of Fe-B and Fe-B (PM) steels, there was an increase in the current after only a few hours of exposure. For Fe-B steel, the current value was almost constant but high and did not vary significantly throughout the test. After approximately 10 h of exposure, activation occurred and a gradual increase in current occurred. The highest current values were achieved by Fe-B (PM) steel. In this case, the increase in the current value was almost instantaneous (about 83 min) and did not increase further. The passive layer of this steel is unstable in the testing environment under oxidizing conditions simulating the cathodic half-cell of the fuel cell. The passive layer instability of the Fe-B and Fe-B (PM) steels probably resulted from chromium-depleted regions in the austenitic matrix adhering to the chromium-rich boride precipitates.

For all materials, a local surface attack was observed after exposure, as shown in Figure 12. For LDX steel and Fe-B steel, visible coloration can be observed in the vicinity of the o-ring. Steel 2205 and Fe-B steel (PM) also showed signs of localized attack below the o-ring. Only on 316L steel could minimal surface changes be observed. Contact resistance measurements were taken after exposure. It can be assumed that the results obtained in contact resistance measurements could be influenced by local attacks on the surface of individual steels. It is necessary to change the experimental setup and eliminate the unfavorable effects caused by the o-ring that influence the final results.

### 3.5. Interfacial Contact Resistance

From the measured contact resistance values on the exposed specimens, it can be concluded that none of the tested steels meet the DOE requirements. For all samples, a contact resistance several times higher than 10 mΩ cm^2^ was achieved. As for the samples tested before potentiostatic polarization, these samples achieved lower contact resistance values because the passive layer was removed from their surface. The contact resistance values obtained at 138 N cm^−2^ are summarized in Table 7, while the graphical interpretation (Figure 13) shows a significant difference between the ground and the exposed condition. From the unexposed samples, the DOE limit was met by steels 316L, 2205, and Fe-B (PM). In the case of exposed specimens, it can be assumed that the value of the contact resistance is closely related to the thickness of the passive layer formed on the steel surface during its previous exposure. The surface roughness itself can affect the contact area and therefore the contact resistance. Using different surface treatment methods (e.g., mechanical grinding, etching, chemical or electrochemical polishing), a more suitable surface condition can be achieved, which can affect the contact resistance [48,75]. Yun observed in his study [76] that the electropolished material exhibited significantly lower contact resistance values than the mechanical grinding material. The coating deposited on electropolished 316 steel is also smoother. On the other hand, the authors observed lower contact resistance values when using the chemical surface treatment on the Fe-Ni-Cr material compared to the electrochemical surface treatment [77].

The contact resistance value can be influenced by the properties of the passive layer, for example, surface conductivity, composition, or thickness. The chemical composition of the tested steels plays a significant role in the formation and chemical composition of the passive layer. The addition of molybdenum and its positive effect on increasing the ionic conductivity of the passive layer was observed in [78]. Chromium plays an important role in any stainless steel, contributing along with molybdenum to the formation of the passive layer. The authors of [79] revealed a significant relationship between the Cr and Mo content of stainless steels and their influence on the reduction in contact resistance. This phenomenon was also observed for the steels tested, with 316L steel achieving better contact resistance values compared to 2205 steel with a higher Cr content. Although a higher manganese content was expected to result in the improved conductivity of the passive layer [80], the LDX steel showed higher values of contact resistance than the 2205 steel.

It should be noted that the surface of the tested steels was affected by crevice corrosion after potentiostatic polarisation, which occurred under the o-ring, as discussed in Section 3.3. The steels tested may have had a non-uniform passive layer thickness, with contact between the thicker part of the passive layer and the carbon paper occurring during the contact resistance measurement.

## 4. Conclusions

Five stainless steels were tested in a simulated fuel cell environment by electrochemical methods.

In accelerated corrosion test measurements:The best results were obtained for austenitic steel 316L and duplex steels 2205 and LDX.Fe-B and Fe-B (PM) steels contain significant amounts of chromium, which is bound in the microstructure in borides, and therefore the surrounding matrix is depleted of chromium and the steel itself loses its protective properties.The results of the accelerated corrosion tests served only for a quick comparison of the tested steels from the corrosion resistance point of view. No conclusions are drawn from them as to the overall suitability of any of the steels tested.

In potentiodynamic tests, according to DOE requirements:The LDX and 2205 duplex steels achieved promising results.The 316L steel achieved approximately similar results to boron-alloyed steels.

In the potentiostatic test with positive polarization:
The results showed that all the materials are prone to crevice corrosion. It cannot be judged responsibly which of the boron-free stainless steels (316L, LDX, 2205) achieved the best results in this experiment.The boron-bearing steels, Fe-B and Fe-B (PM), exhibited much higher corrosion current values above the DOE limit as a result of the limited passivability and stability of passivation resulting from chromium depletion by chromium boride precipitates.

In the results of contact resistance measurements:Each of the tested steels exceeded the DOE limits after potentiostatic testing.

Based on the results, it is evident that stainless steels, including those with boron or manganese addition, do not provide sufficient corrosion resistance in the PEMFC environment if no additional surface treatment is applied. Therefore, the application of a surface treatment or the application of a protective coating is recommended in order to increase the corrosion resistance and reduce the contact resistance of stainless steels.

## Figures and Tables

**Figure 1 materials-15-06557-f001:**
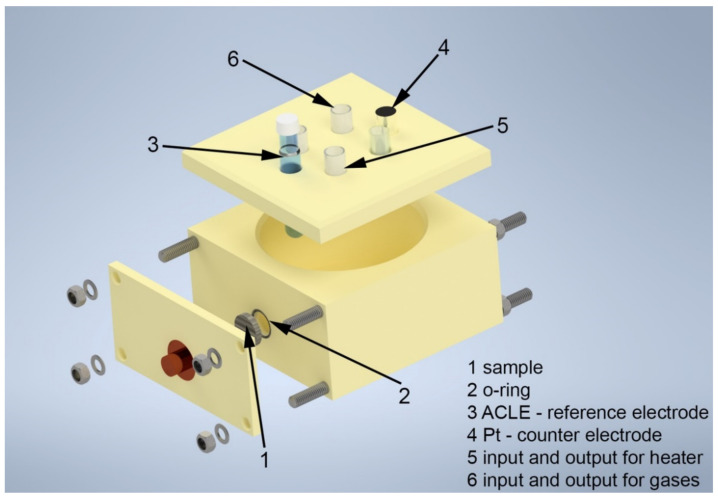
Three-dimensional model of corrosion cell used for all electrochemical measurements.

**Figure 2 materials-15-06557-f002:**
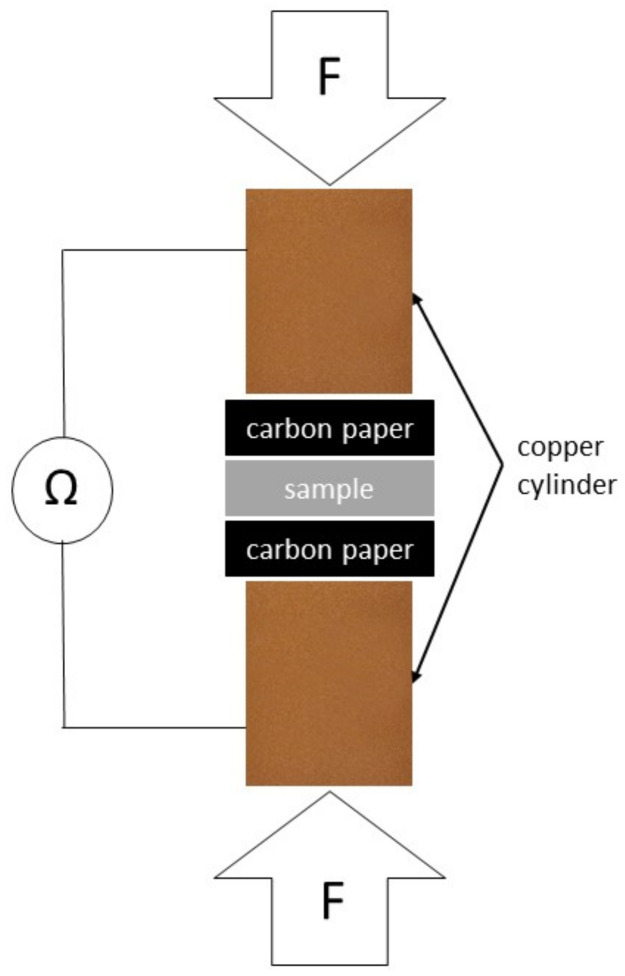
Scheme of a contact resistance measurement.

**Figure 3 materials-15-06557-f003:**
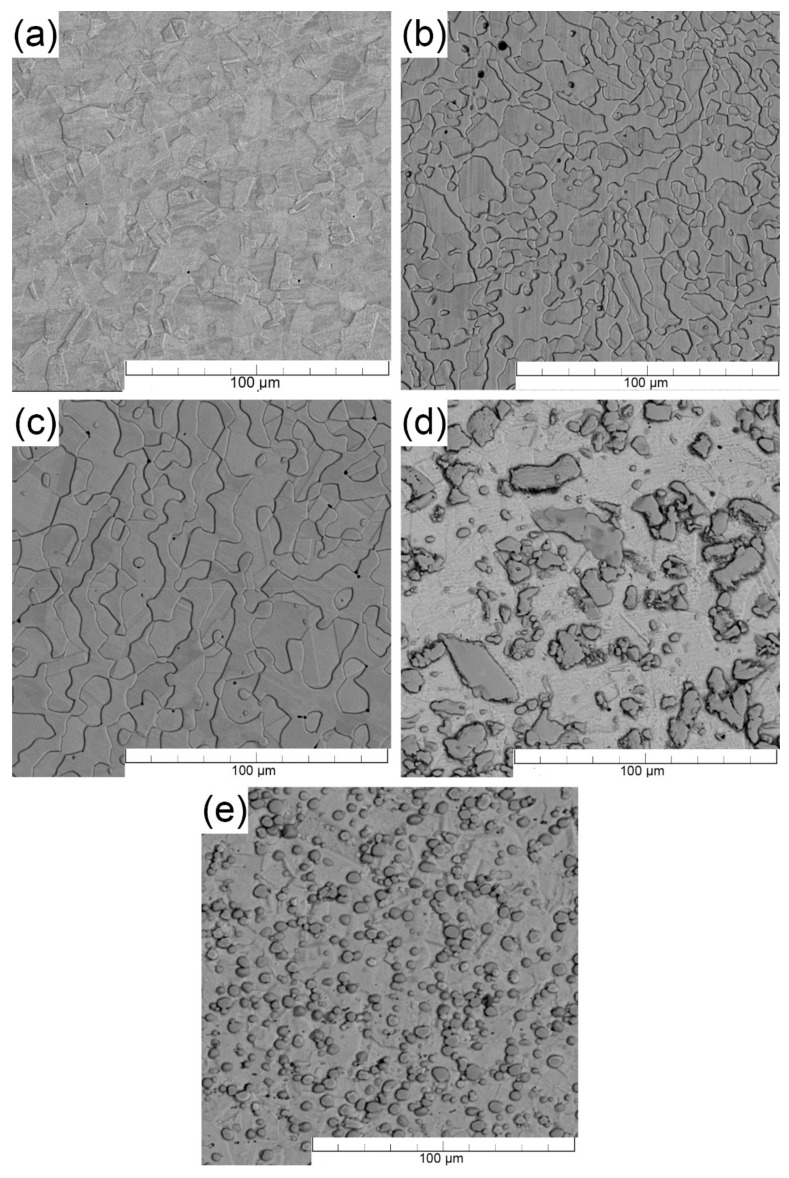
Microstructures of steels: (**a**) 316L, (**b**) LDX, (**c**) 2205, (**d**) Fe-B, (**e**) Fe-B (PM).

**Figure 4 materials-15-06557-f004:**
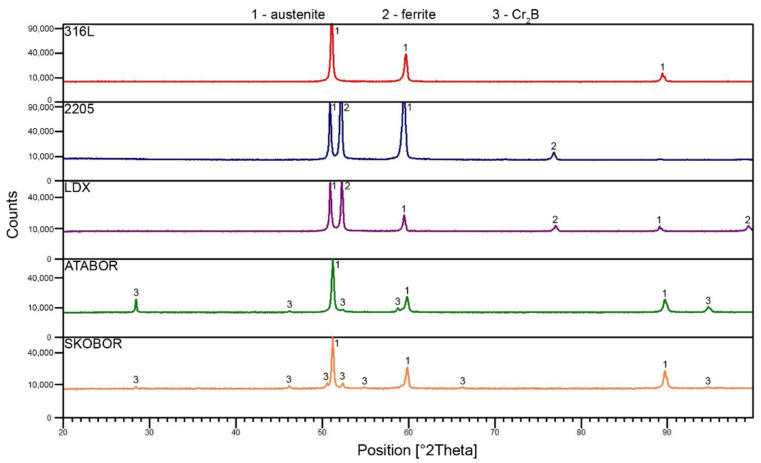
X-ray diffraction patterns of the studied stainless steels.

**Figure 5 materials-15-06557-f005:**
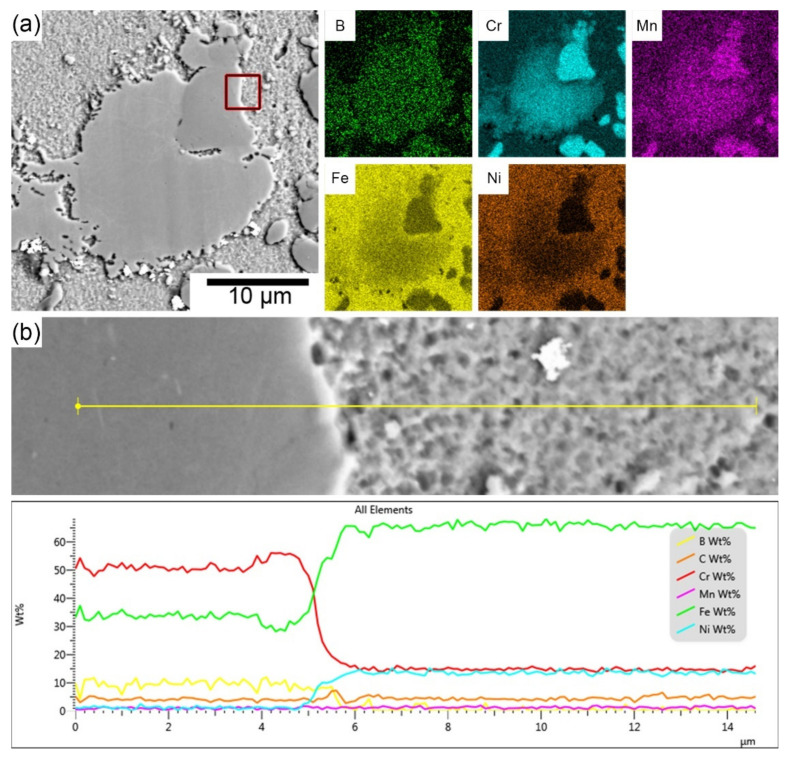
(**a**) SEM elements distribution maps of the Fe-B stainless steel, (**b**) line scan through the boride particle and austenitic matrix in the Fe-B stainless steel.

**Figure 6 materials-15-06557-f006:**
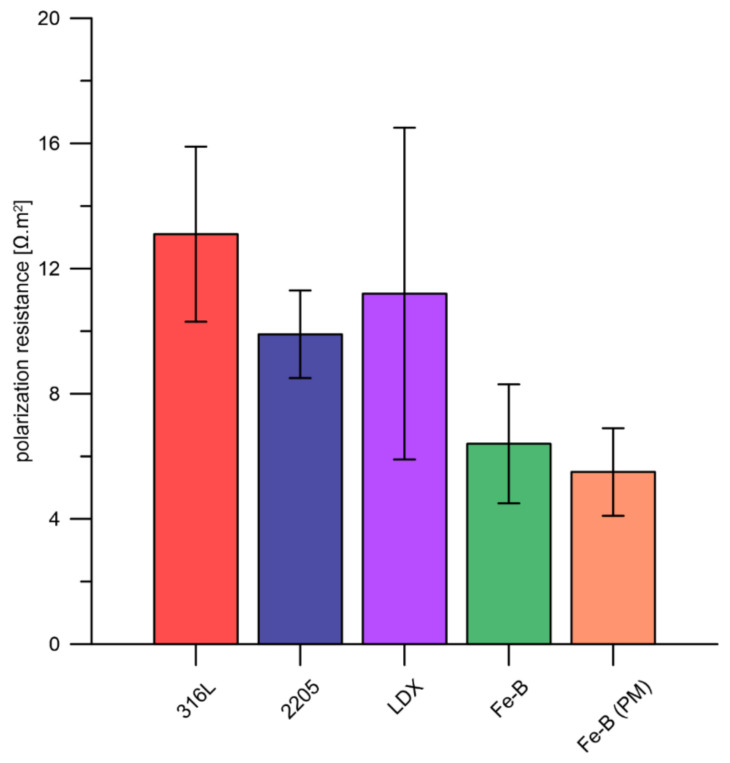
Results of polarization resistance measurement in 0.1 ppm F^−^, pH = 3, temperature 80 °C (average values from up to 5 replicates supplemented with standard deviation error bars).

**Figure 7 materials-15-06557-f007:**
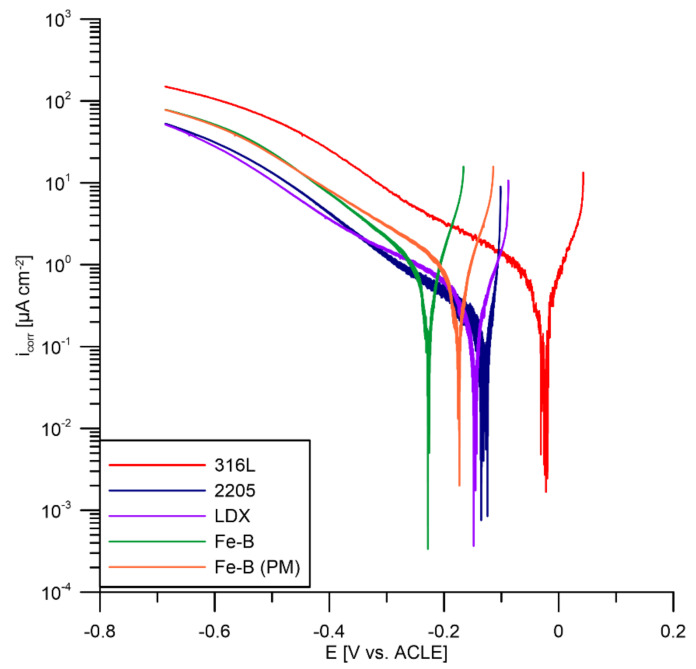
Cathodic parts of potentiodynamic curves for tested steels in PEMFCs.

**Figure 8 materials-15-06557-f008:**
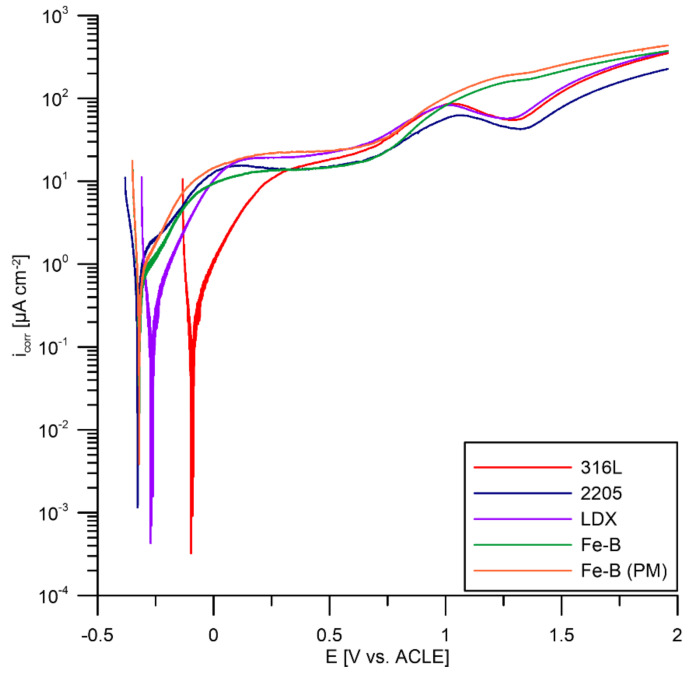
Anodic parts of potentiodynamic curves for tested steels in PEMFCs.

**Figure 9 materials-15-06557-f009:**
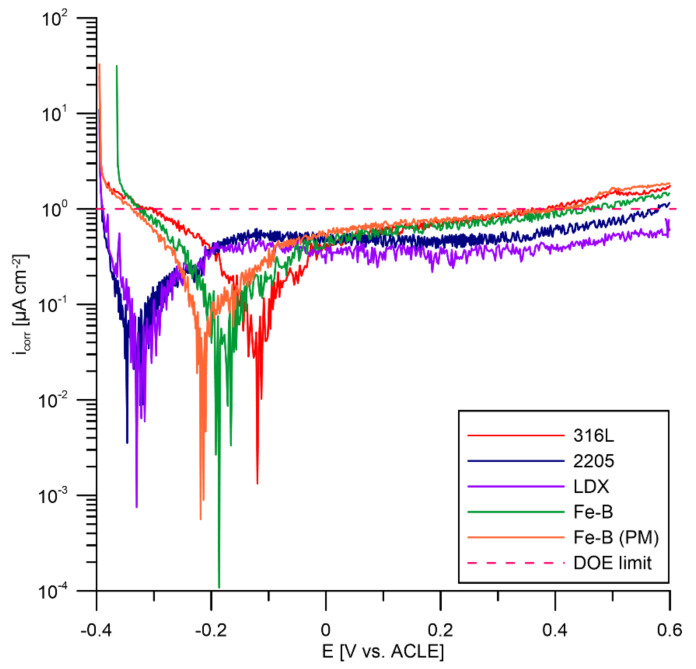
Potentiodynamic polarization in the deaerated environment of PEMFCs.

**Figure 10 materials-15-06557-f010:**
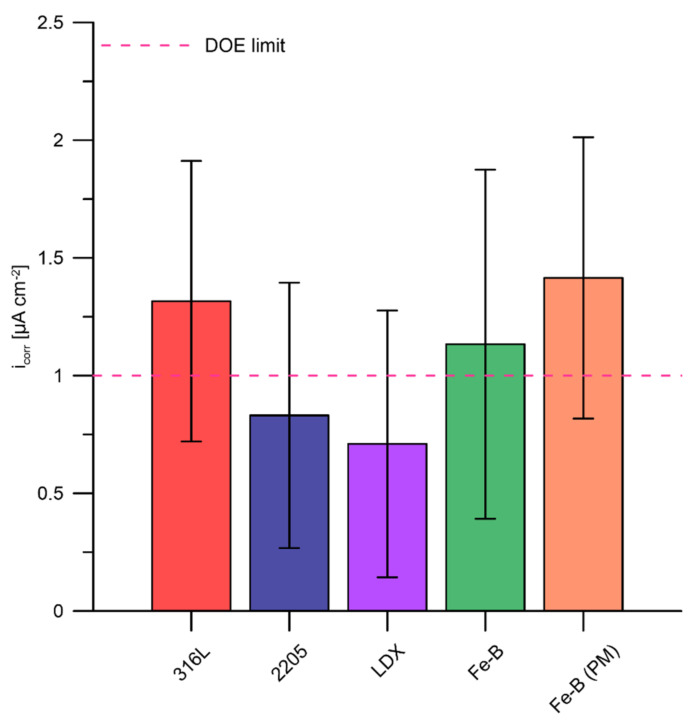
Current density (µA cm^−2^) at a potential of 0.6 V vs. ACLE for test steels (average values from up to 5 replicates supplemented with standard deviation error bars).

**Figure 11 materials-15-06557-f011:**
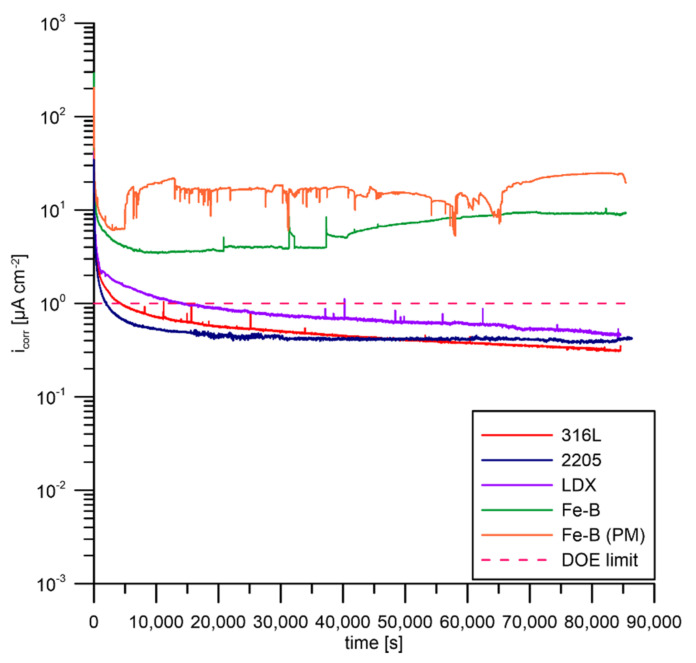
Potentiostatic polarization according to DOE requirements.

**Figure 12 materials-15-06557-f012:**
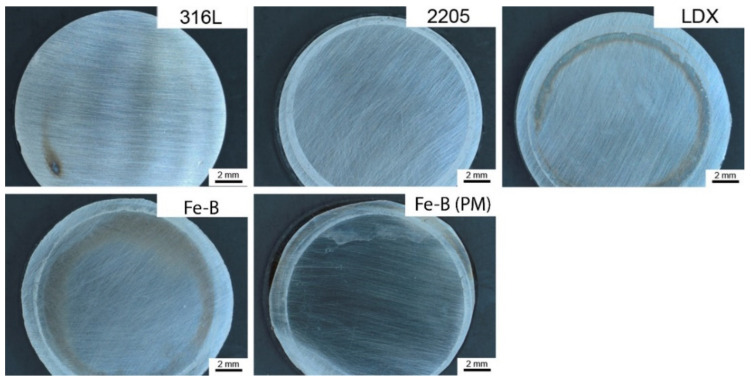
Surface of samples after the potentiostatic polarization.

**Figure 13 materials-15-06557-f013:**
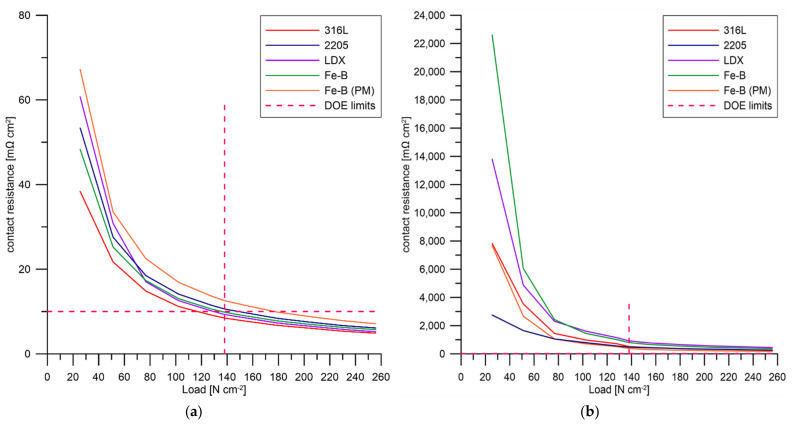
Contact resistance. (**a**) Ground surface, (**b**) exposed surface.

**Table 1 materials-15-06557-t001:** Elongation values for the most commonly tested materials [40,41,42].

	304 1.4301	316L 1.4404	904L 1.4539	254SMO 1.4547	2205 1.4462	LDX 1.4162	Pure Ti Grade 2	DOE Requirement
A (%)	≈55	≈55	≈50	≈50	≈35	≈38	≈20	40

**Table 2 materials-15-06557-t002:** The DOE requirements for 2020 [59].

Property	Target	Notes
Cost	<3 $ kW^−1^	2002 dollars, 500,000 stacks per year
Corrosion resistance (anode)	<1 µA cm^−2^ no active peak	pH 3, 0.1 ppm HF, 80 °C, Ar purge Potentiodynamic test −0.4 V to + 0.6 V (ACLE), 0.1 mV s^−1^
Corrosion resistance (cathode)	<1 µA cm^−2^	pH 3, 0.1 ppm HF, 80 °C, aerated Potentiostatic test (>24 h) 0.6 V (ACLE), i_passive_ < 1 µA cm^−2^
Electrical conductivity	>100 S cm^−1^	-
Areal specific resistance	<0.01 Ω cm^2^	Including contact resistance at 138 N cm^−2^
Hydrogen permeability	<1.3 × 10^−14^ cm^3^	ASTM D1434, 80 °C, 3 atm, 100% RH
Flexural strength	>25 MPa	ASTM D790-10
Forming elongation	40%	ASTM E8M-01

**Table 3 materials-15-06557-t003:** Chemical composite of stainless steels tested (in wt.%).

	C	Cr	Ni	Mo	Cu	B	Mn
FeCr18Ni13Mo3 (316L)	0.03	17.4	14.4	2.8	-	-	1.8
FeCr21Ni5Mo3Mn2 (2205)	0.04	22.6	5.6	2.6	-	-	1.6
FeCr21Ni2Mo0.5Cu0.5Mn4 (LDX)	0.03	21.5	1.5	0.2	0.2	-	4.7
FeCr18Ni13B1Mn2 (Fe-B)	0.03	19.1	12.7	-	-	1.2	1.1
FeCr19Ni13B1Mn2 (Fe-B-PM)	0.04	19.2	13.3	-	-	1.2	1.5

**Table 4 materials-15-06557-t004:** Values of E_corr_ for all tested materials in the PEMFC model environment.

Material	316L	LDX	2205	Fe-B	Fe-B (PM)
E_corr_ (mV vs. ACLE)	−159 ± 66	−301 ± 93	−282 ± 49	−313 ± 71	−259 ± 28

**Table 5 materials-15-06557-t005:** Values of E_corr_, i_corr_, and β_c_ determined from the cathodic parts of potentiodynamic curves for all tested materials.

Specimen	316L	2205	LDX	Fe-B	Fe-B (PM)
E_corr_ (V vs. ACLE)	−0.022	−0.135	−0.148	−0.228	−0.173
i_corr_ (μA cm^−2^)	0.245	0.177	0.371	0.815	0.800
β_c_ (V per decade)	0.251	0.198	0.251	0.171	0.214

**Table 6 materials-15-06557-t006:** Values of E_corr_, i_corr_, and β_a_ determined from the anodic parts of potentiodynamic curves for all tested materials.

Specimen	316L	2205	LDX	Fe-B	Fe-B (PM)
E_corr_ (V vs. ACLE)	−0.097	−0.328	−0.272	−0.320	−0.320
i_corr_ (μA cm^−2^)	0.482	1.051	0.192	0.816	0.963
β_a_ (V per decade)	0.237	0.261	0.192	0.306	0.205

**Table 7 materials-15-06557-t007:** Summarized values of contact resistance (in mΩ cm^2^).

Specimen	316L	2205	LDX	Fe-B	Fe-B (PM)
Exposed	454	438	782	670	309
Ground	8.5	10.6	11.9	9.9	12.6

## Data Availability

The raw/processed data will be provided as request.

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
