# Peer review of "Corrosion Properties of Boron- and Manganese-Alloyed Stainless Steels as a Material for the Bipolar Plates of PEM Fuel Cells"

_materials, 2022, doi:10.3390/ma15196557_

Round 1
Reviewer 1 Report (Previous Reviewer 1)
In this manuscript, the authors reported corrosion properties of five stainless steels were tested in a simulated fuel cell environment by electrochemical methods. Overall, the content of the paper is relatively substantial. But there are still some problems in the results and some statements is not rigorous enough, which greatly lowers the quality of the paper. In addition, the analysis and discussion also are weak as a whole. Therefore, the reviewer suggests major revision the manuscript in its present state, and the following points should be considered by the authors:
1. The language needs to be carefully polished throughout the paper. In addition, the abstract and conclusions are too complicated and should be simplified.
2. The research progress regarding PEMFC environment of corrosion properties of different Stainless steels is not sufficiently reviewed in the Introduction section. In fact, there are some reports, which should be respected.
3. Section 3.1, the authors present formed particles are mainly enriched in Cr and Mn, Boron bound to chromium forms (Fe, Cr)2B type borides and M23(C, B)6 carbide-borides. The above speculation is not rigorous. The content of Mn in the products is relatively high. Correspondingly, the XRD analysis results should be checked or More detailed TEM microstructures are needed.
4. Check the data in Table 7, some values clearly did not match Figure 13.
5. Section 3.5, the authors present the surface roughness itself can affect the contact area and therefore the contact resistance. More explanations and discussions regarding the reasons should be provided.
6. Section 3.5, the authors present the contact resistance value can be influenced by the properties of the passive layer, but the experiments in this article were not sufficient to support this view.
Author Response
Dear reviewer,
thank you for your valuable comments.
- The abstract and conclusions have been modified. The language will be edited afterwards, the paper has been sent for language correction.
- This publication deals primarily with corrosion behavior in the context of basic research. A great review of the material has been done by my colleagues (Bohackova, T., Ludvik, .J. and Kouril. M., Metallic Material Selection and Prospective Surface Treatments for Proton Exchange Membrane Fuel Cell Bipolar Plates—A Review. Materials, 2021. 14, DOI: 10.3390/ma14102682.). But I would ask you to be more specific which papers were not accepted by this paper.
- The assertions in section 3.1 were supported by additional literature. Anyway, thank you for your comment and the suggestion that these TEM microstructures should appear in the next publication.
- Thank you for this reminder, figure 13 has been corrected.
- Additional explanations regarding surface roughness have been added and have also been supported by the literature.
- Thank you for this comment. Yes, the experiments in this paper were not performed to support this view. The authors relied on literature such as Iversen, A.K., Stainless steels in bipolar plates—Surface resistive properties of corrosion resistant steel grades during current loads. Corrosion Science, 2006. 48(5): p. 1036-1058.

Reviewer 2 Report (Previous Reviewer 3)
The study submitted by Tomáš Lovaši presents the analysis of boron and manganese alloyed stainless steels for potential usage in bipolar plates for PEM fuel cells. After re-analysing the paper, the authors have resolved the raised issues, and I recommend publication.
Author Response
Dear reviewer,
thank you for your favorable review of our paper.
Reviewer 3 Report (Previous Reviewer 4)
The paper has been revised and may be accepted in present form
Author Response
Dear reviewer,
thank you for your favorable review of our paper.
This manuscript is a resubmission of an earlier submission. The following is a list of the peer review reports and author responses from that submission.
Round 1
Reviewer 1 Report
This paper has investigated corrosion behavior and interfacial contact resistance of five stainless steels. The results showed that none of them provide sufficient corrosion resistance in the PEMFC environment, and all the tested steels exceeded the limits after potentiostatic testing. The present paper reached a conclusion that a surface treatment or a protective coating is needed to the bipolar plates of PEMFC. The originality of this paper is insufficient.
As we all know, the thickness of stainless steel bipolar plates of PEMFC is only 0.1mm. And the elongation needs to reach 40%,as mentioned in the manuscript. Only 316L can meet the two requirements among the tested materials. It is difficult to manufacture bipolar plates for the low ductility of duplex stainless steel and the brittleness of Fe-B steel. Therefore, the materials suitable for stainless steel bipolar plates need to be further explored, for example, super ferritic stainless steel.
Author Response
Dear reviewer,
thank you for your valuable comments. This publication deals primarily with corrosion behavior in the context of basic research and therefore does not address all DOE conditions. One of the main factors is the shape of the bipolar board – flow field design. It is possible that future designs will not require such a large deformation of the material. The actual process of making thin bipolar plates is also yet to be investigated. Improvements in the design of bipolar plates can help to achieve the set goals.
Song, Y.; Zhang, C.; Ling, C.-Y.; Han, M.; Yong, R.-Y.; Sun, D.Chen, J. Review on current research of materials, fabrication and application for bipolar plate in proton exchange membrane fuel cell. International Journal of Hydrogen Energy 2020, 45, 29832-29847. https://doi.org/10.1016/j.ijhydene.2019.07.231
Reviewer 2 Report
The work is interesting and well written but some major modifications are required before publication. Below my specific comments:
-Please add in the abstract section some main test results (electrochemical tests and contact resistance measurements).
-Please add in the introduction section (line 57–line 59) some recent reference where stainless steels with good mechanical properties, good electric and heat conductivity.
-In the “electrochemical measurements”(line 115–line 119), “A platinum plate with an area of 0.5 cm2 was used as a reference electrode.” In my opinion, The potential of the platinum electrode is not stable, how to eliminate this error or what literature is referenced in this work, please explain?
-In the “accelerated corrosion tests”(line 138–line 139), “The cathodic part of the polarization curve was measured in the potential range of +100 mV vs. Ecorr to -1.2 V vs. Eref with a scan rate of 5 mV s-1. The anodic portion of the curve was measured at the same scan rate over a potential range of -100 mV vs. Ecorr to 1.5 V vs. Eref.” It is worth noting that potential scan rate has an important role in order to minimize the effects of distortion in Tafel slopes and corrosion current density analyses, as previously reported “[AA-BB]”. Excessive scan rate will cause large measurement error, please check the experimental results again.
[AA] Zhang X.L., Jiang Zh.H., Yao Zh.P, Song Y., Wu Zh.D. Effects of scan rate on the potentiodynamic polarization curve obtained to determine the Tafel slopes and corrosion current density. Corrosion Science. 2009, 51: 581-587.
[BB] E. McCafferty. Validation of corrosion rates measured by the Tafel extrapolation method. Corrosion Science 47 (2005) 3202-3215.
-Please add enlarged views of local surface attack in Figure 11.
Author Response
Dear reviewer,
Thank you for your valuable comments and positive review.
- The main results discussed in this publication have been added to the abstract.
- In my opinion, such steel doesn't even exist. There is a incorrect combination of words in the sentence. I've reformulated the sentence to its correct meaning.
- A more specific explanation of the choice of the reference electrode is described in more detail in the article (line 123-130) All values were converted to ACLE according to a single measurement at the appropriate time. It is clear from the average values that the potentials are not significantly different.
- Definitely, the reviewer’s comment is correct. Relatively high scan rate was used just for preliminary comparison of corrosion behavior of the tested materials. Much lower scan rate was used for corrosion testing according to the DOE recommendations as it is presented in the paper.
- Unfortunately, the samples were destroyed during the contact resistance measurement and it is not possible to acquire these images.
Reviewer 3 Report
The study submitted by Tomáš Lovaši presents the analysis of boron and manganese alloyed stainless steels for potential usage in bipolar plates for PEM fuel cells. Materials and investigations are properly conducted, explaining microstructure and corrosion tests. Also compare the investigation with DOE standards.
The research try to find an alternative solution for bipolar plates of PEM fuel cells, according to DOE standards. The topic is original, and try to solve some issues of existing materials. As innovation it add the use of some steel materials, but this seems rather limited as materials. But represents an interesting approach. The study is well conducted, the authors should only consider to interpret all their data: X-ray diffraction. Also SEM analysis should be larger included in the study. The conclusions are consistent with the evidence and arguments presented and they address the main question posed. References are appropriate.
After analysing the paper, I recommend publication after resolving some minor remakes:
- Figure 5 is not introduced in text and not discussed.
- Why the author didn’t not include a SEM image before and after potentiostatic polarization tests?
Author Response
Dear reviewer,
Thank you for your valuable comments and positive review.
- Figure 5 has been newly introduced in the publication and subsequently discussed.
- SEM images before and after potentiodynamic polarization were unfortunately not taken. Anyway, thank you for your comment and the suggestion that these images should appear in the next publication.
Reviewer 4 Report
The paper present the test results of a number of possible stainless steel in a simulated fuel cell environment for using as a material for bipolar plates of PEM fuel cells. The study is an interesting and useful piece of work therefore it is worth publishing. The article was written in a simply and flowing manner which is easy to understand. Paragraph 1 of Introduction section is too long and contains several main points. Therefore, it is advisable to divide it into several paragraphs.
Author Response
Dear reviewer,
Thank you for your valuable comments and positive review. According to your comment, the paragraph has been split.
Round 2
Reviewer 1 Report
This paper studies the feasibility of five stainless steels as a material for Bipolar Plates of PEM Fuel Cells. The corrosion behavior was tested in a simulated fuel cell environment. Also, the formability of materials must be considered. Otherwise, the research will lose its significance.
Reviewer 2 Report
My previous comments were not well answered.
The stability of the reference electrode in the three-electrode system is very important. In my experience, the platinum electrode as the reference electrode has a large potential drift in the test solution. In your study, “It is clear from the average values that the potentials are not significantly different.” I don't think this argument is convincing, please provide more experimental data and explain.